# Albumin as a Biomaterial and Therapeutic Agent in Regenerative Medicine

**DOI:** 10.3390/ijms231810557

**Published:** 2022-09-12

**Authors:** Olga Kuten Pella, István Hornyák, Dénes Horváthy, Eszter Fodor, Stefan Nehrer, Zsombor Lacza

**Affiliations:** 1Orthosera GmbH, 3500 Krems an der Donau, Austria; 2Institute of Translational Medicine, Semmelweis University, 1094 Budapest, Hungary; 3Department of Interventional Radiology, Semmelweis University, 1122 Budapest, Hungary; 4Institute for Sports and Health Sciences, Hungarian University of Sports Science, 1123 Budapest, Hungary; 5Center for Regenerative Medicine, Danube University Krems, 3500 Krems an der Donau, Austria

**Keywords:** albumin, serum albumin, bone allograft, scaffolds, regenerative medicine

## Abstract

Albumin is a constitutional plasma protein, with well-known biological functions, e.g., a nutrient for stem cells in culture. However, albumin is underutilized as a biomaterial in regenerative medicine. This review summarizes the advanced therapeutic uses of albumin, focusing on novel compositions that take advantage of the excellent regenerative potential of this protein. Albumin coating can be used for enhancing the biocompatibility of various types of implants, such as bone grafts or sutures. Albumin is mainly known as an anti-attachment protein; however, using it on implantable surfaces is just the opposite: it enhances stem cell adhesion and proliferation. The anticoagulant, antimicrobial and anti-inflammatory properties of albumin allow fine-tuning of the biological reaction to implantable tissue-engineering constructs. Another potential use is combining albumin with natural or synthetic materials that results in novel composites suitable for cardiac, neural, hard and soft tissue engineering. Recent advances in materials have made it possible to electrospin the globular albumin protein, opening up new possibilities for albumin-based scaffolds for cell therapy. Several described technologies have already entered the clinical phase, making good use of the excellent biological, but also regulatory, manufacturing and clinical features of serum albumin.

## 1. Introduction

Albumin is the most abundant protein in plasma and plays important metabolic roles, such as regulating the oncotic pressure, binding and transporting various molecules, scavenging free radicals and modulating the immune response and blood coagulation [1]. There are different types of albumin, such as human serum albumin (HSA); animal serum albumin: bovine serum albumin (BSA), rat serum, etc.; albumin from eggs (ovalbumin), milk (lactalbumin) and plants sources such as soy; and grains [2,3] (Figure 1). Different types of albumin demonstrate similarities; however, some differences were also reported (Table 1).

Mammalian albumins, as well as OVA, are α-helical globular and water-soluble proteins and contain hydrophilic and hydrophobic sites with an acidic characteristic [3,27]. Albumin is a very stable and highly soluble protein tolerant of high temperatures. Thanks to its disulfide bonds and sulfhydryl groups, which allow interactions with organic and inorganic ligands, albumin can be described as chemically attractive. Albumin is among the most studied of all proteins; thus, it is exploited in various biotechnological applications: as a drug, theranostic agent, biomaterial, in vitro cell supplementation, biosensor, contrast agent, etc. [2,3,7,28]. 

Regenerative medicine supports natural healing processes by using autologous and allogeneic cells, biomaterials, growth factors, gene manipulation or combinations of these elements [29]. In the present work, we review how albumin can be utilized as a biomaterial and therapeutic agent in the biomedical field. It is commonly understood that the clinical usability of a biomaterial depends on its biocompatibility, i.e., that it does not trigger any side effects such as irritation, inflammation, cytotoxicity, genotoxicity or mutagenicity [30,31,32]. Biocompatibility also means that the material possesses its designed function for therapy and maintains a regenerative response to the surrounding tissue environment [31]. Since albumin is a main mammalian blood component, it has excellent biocompatibility [3,5,26]; however, despite its wide clinical application and recorded use in tissue engineering, albumin is still not commonly used in regenerative medicine. Therefore, this review article focuses on the current findings about the use of albumin as a biomaterial or a local therapeutic agent, especially in regenerative therapies and novel tissue engineering solutions.

## 2. Albumin as a Biomaterial

A range of biomaterials is used in regenerative medicine. Metal-based biomaterials such as gold, platinum, titanium, steel, etc. are suitable due to their inertness and structural functions; however, their surfaces do not possess bioactivity [31,33]. Ceramics and bioglasses characterized by high biocompatibility and good mechanical properties find application in dentistry and bone regeneration despite their poor plasticity [31,34]. Synthetic polymers (PU, PP, PLA, PEG, PMMA, etc.) are used for the production of scaffolds, prostheses, implants, medical devices and even contact lenses [31]. They have good mechanical properties, slow degradation rates and adjustable architecture but do not attract cell attachment and spreading. Some novel solutions for enhancing the biocompatibility include the use of natural materials, decellularized organs, blood-derived products, primary cells or stem cells. Natural polymers include proteins of the extracellular matrix (ECM) such as collagen, fibrinogen or hyaluronic acid, which promote cell adhesion and have superior biocompatibility, good biological activity, low immunogenicity and low cytotoxicity from their degradation products. However, in the case of natural biomaterials, there is always a concern regarding their stability and degradation rate [31,35,36,37,38,39]. Albumin is an example of a blood-derived product with a potential for autologous or allogeneic tissue engineering [40], and some reports show its successful use in coating, as well as scaffolds and hydrogels fabrication. In the following subsections, we will discuss the examples of combining albumin with different materials or its direct use as a biomaterial for medical applications.

### 2.1. Coating

Biomaterials are often coated with natural polymers such as dextran, collagen, chitosan or serum albumin [41,42,43]. There are two different approaches for coating biomaterials: permanent or temporary. For permanent medical devices, it is important that their surface is passive, can resist for a longer time and take over the function of damaged tissue—albumin with its high bioactivity and solubility is hardly suitable for the type of permanent coating. In contrast, these exact same properties make albumin optimal for temporary coating other implantable biomaterials, thereby increasing their biocompatibility. The current bioengineering solutions are expected to stimulate self-regeneration of the tissue by attracting cell attachment, proliferation and even differentiation [44]. Several approaches for albumin immobilization on the biomaterial surface, such as physical adsorption, crosslinking with different reagents and photoactivable albumin, are in use; however, it was shown that a simple lyophilization is enough to coat bone allografts with a biologically significant amount of serum albumin (Figure 2) [45]. Albumin coatings were shown to improve the bio- and immune compatibility, tissue formation, corrosion resistance and antibacterial and anticoagulant properties of materials in several studies [26,42,43,46,47,48,49,50]. Most studies described albumin as a cell adhesion-inhibiting protein on inert surfaces, but it is a very potent cell adhesive in more physiological scaffolds such as human bone allografts [51,52,53]. 

### 2.2. Anticoagulant Properties

It is important to understand the complex adsorption processes on the biomaterial surface when it is placed in vivo and comes into contact with body fluids. Protein adsorption is one of the first biological events after implanting a foreign material into a living organism [54]. Depending on the physical and chemical properties of the implanted biomaterial, it can attract different proteins, platelets, immune cells, fibroblasts, stem cells, etc. [54,55,56]. Albumin was one of the first proteins to be used as a coating to prevent surface-induced platelet activation [57,58] Albumin precoating provides a thin protein layer that increases the hydrophilicity of the surface and prevents a biological response after contact with the blood of an otherwise hydrophobic material; this method is called “albumin passivation” [50,58]. Materials with adsorbed native albumin were shown to reduce the number of adherent platelets and their activation on the surface; however, when the albumin structure was changed by crosslinking, platelets were able to adhere and activate completely to the modified albumin layer [45]. Studies showed that platelets can adhere to adsorbed albumin only if albumin undergoes more than a 34% loss in its α-helical content [59,60]. In the past, the arterial vascular prosthesis material Dacron was used for artery replacement; however, its use was problematic due to thrombogenic reactions. Attempts with using collagen and gelatin did not decrease the thrombogenicity. The grafts were further immersed into a buffered solution of albumin and glutaraldehyde, which produced a crosslinked albumin coating. Albumin diminished coagulation activation and fibrinopeptide A formation, as well as leucocyte and platelet adhesion; therefore, there was a significant improvement in the short-term blood compatibility of Dacron [61]. Several current commercial products take advantage of albumin coatings, e.g., perfusion systems, catheters and cannula; extracorporeal life support systems and human bone allografts [47,62,63,64,65,66,67,68,69]. There are also more complex approaches that combine albumin coating and novel solutions. Abraham et al. fused HSA with ectonucleoside triphosphate diphosphohydrolase-1 (CD39) and used this HSA-CD39 fusion protein as a coating with antithrombotic and anti-inflammatory activities for medical devices. CD39 in this complex is intended to shift a proinflammatory environment to an anti-inflammatory status, and this might prevent the rejection of implants. Reduced platelet adhesion, minimal platelet aggregation and blood coagulation were observed, therefore providing an opportunity to reduce the thrombotic and inflammatory reactions in medical devices coated with HSA-CD93 [70].

### 2.3. Antibacterial Properties

Another advantage of the albumin coating is its antimicrobial properties. An et al. evaluated the effects of albumin coating on titanium implants in preventing infection in rabbits. All the implants were exposed to a suspension of *S. epidermidis* before implantation. They observed a significant reduction in the infection rate, from 62% in rabbits with noncoated implants to 27% in rabbits with albumin-coated implants [71]. Another study monitored the albumin coating and the two of the most commonly implicated pathogens that result in biofilm formation, *S. aureus* and *P. aeruginosa*, that can adhere to the surface of an implanted material [72,73]. When a biofilm is formed, bacteria can easily evade the host immune response and become resistant to treatment, which, in the end, might result in implant failure [73,74]. Significantly fewer bacteria adhered to the HSA-coated titanium plate than to the uncoated surfaces. The binding of *S. aureus* was inhibited significantly from 82% to 95% and, of *P. aeruginosa*, from 29% to 37% on the HSA-coated surface [73]. Sun et al. combined small-molecule gold nanoparticles with 4,6-Diamino-2-pyrimidinethiol (Au-DAPT) and BSA to obtain the antimicrobial conjugate Au-DAPT-BSA with antibacterial activity against several Gram-negative and Gram-positive bacteria, including drug-resistant bacteria and *S. aureus* and *P. aeruginosa.* In vitro and in vivo tests showed excellent antibacterial efficacy, no cytotoxicity to mammalian cells and that Au-DAPT-BSA-based therapy can be helpful in wound healing and skin infections [75]. Cometta et al. highlighted an infection problem associated with breast implant surgeries. They proposed biodegradable medical-grade polycaprolactone (mPCL) scaffolds with HSA immobilized on their surfaces as an alternative to silicon implants. HSA was crosslinked with tannic acid (TA) and tested for 7 days. The 1%HSA/10%TA- and 5%HSA/1%TA-coated scaffolds were able to reduce *S. aureus* colonization on the mPCL surface by 99.8 ± 0.1% and 98.8 ± 0.6% when compared to the noncoated control scaffolds [76].

### 2.4. Albumin as Cell Adhesion Protein

Albumin is generally known as a cell adhesion-inhibiting protein [51,52,53]. However, multiple in vitro and in vivo studies proved that, despite its antiadhesive properties on plastic surfaces, albumin coating supports mammalian cell growth and tissue formation [47,49,63,64,66,67,69,77]. Weszl et al. compared the adherence and proliferation of mesenchymal stem cells (MSCs) on albumin-coated and uncoated lyophilized human bone allografts, hydroxyapatite and lyophilized bovine bone, as well as on allografts with different types of coatings. Freeze-dried albumin coating outperformed the liquid coatings, as well as freeze-dried fibronectin and collagen I, in cell attachment and proliferation [66]. An analysis of macro-, micro- and nanostructures of grafts showed amorphous chip-like structures of freeze-dried albumin coating (Figure 1). In contrast to 2D monolayer cell cultures, the attached MSCs did not cover the surface but rather spanned the pores, with only filaments touching the graft surfaces. Additionally, albumin-coated grafts implanted in vivo into bone defects resulted in better graft integration. Interestingly, albumin coating was effective only on human bone materials but not on hydroxyapatite or bovine bone scaffolds. The positive effects of albumin coating on cell adhesion and proliferation were observed, probably due to the enhanced water absorption by freeze-dried albumin that creates an optimal microenvironment for MSCs with high local albumin contents. Serum is one of the standard supplementations for cell cultivation; thus, this microenvironment nurtures MSCs. Cells might easily regain their metabolic activity and start to deposit their own ECM, which further supports attachment and proliferation. The human bone structure and pore size probably play a crucial role in this mechanism [66]. Furthermore, Skaliczki et al. also observed enhanced albumin-coated bone graft remodeling in vivo when compared to uncoated grafts [64], and Horvathy et al. presented results with a significantly reduced healing period in critical-sized bone defects with the formation of mechanically stronger bone with demineralized bone matrices (DBM) and serum albumin [49]. A clinical study with 10 patients who required large structural allografts in hip and knee prosthesis revisions showed good radiography, and the SPECT-CT follow-up showed good biocompatibility, active tissue remodeling and no complications after 12 months [77]. Consequently, it was demonstrated in several double-blinded clinical studies that donor site pain after bone-patellar tendon-bone surgery or extraction socket filling is significantly reduced if the bone buildup is augmented by serum albumin-coated bone allografts [67,69].

Albumin coating on bone allografts was further tested in vivo by creating bone defects on the parietal bones of aging female rats. BoneAlbumin and a noncoated demineralized bone matrix (DBM) were used to fill in the bone defects, and the graft integration with tissue was determined by computed tomography (CT), microCT and mechanical testing. In vivo CT and ex vivo microCT measurements showed faster and stronger bone formation and had a two-times higher fracture force compared to the uncoated bone grafts. After in vitro incubation with MSCs, the albumin-coated grafts attracted approximately twice as many cells as the uncoated grafts [47]. Simonffy et al. compared BoneAlbumin to bovine xenografts in mandibular third molar extraction sockets in a study with 24 patients in a double-blinded study design. Albumin-coated grafts had the lowest level of postoperative pain, and after 6 and 12 weeks, there were signs of tissue remodeling, while uncoated xenografts were still demarcated from the host bone. One-year CBCT images showed complete remodeling and integration, with a natural trabecular structure [69]. The recruitment of cells such as MSCs into the defect region supports tissue formation; however, the question arises if the inflammatory cells are also attracted by albumin coating [63]. Therefore, Mijiritsky et al. investigated interactions in the coculture of monocytes and stem cells on albumin-coated bone grafts and bovine xenograft granules. One of the observations was that monocytes have the ability to degrade uncoated bovine bone granules and that albumin coating protects grafts from degradation. There was also a significant decrease in the reactive oxygen species (ROS) and reactive nitrogen species (RNS) levels in the albumin-coated group. Furthermore, the results of the mitochondrial energy metabolism gene expression suggest that albumin coating relieves the gene expression in inflamed and noninflamed conditions, resulting in a lower amount of differentially expressed genes compared to cells seeded on uncoated xenografts. The cytokine analysis revealed that culturing stem cells and monocytes on albumin-coated grafts resulted in increased levels of HGF cytokines, which is important for tissue repair processes and anti-inflammatory cytokines PGE-2 and IL-10 versus uncoated xenografts. This study shows that albumin, alongside all the previously mentioned features, might also have immunomodulatory functions that are important in healing and tissue regeneration [63].

In addition to numerous publications on bone allografts coated with albumin, bone is not the only tissue used in regenerative medicine that was combined with albumin. For example, polyfilament absorbable sutures were coated with fibronectin, poly-L-lysine and albumin and were seeded with MSCs. After 48 h, albumin-coated sutures had the highest cell number in vitro, and after implantation of the sutures to triceps surae muscle, cells started to detach from the albumin-coated surface and migrate into the injured tissue. Therefore, albumin-coated biomaterials have the potential for soft tissue regeneration [48]. An interesting solution was presented by Wang et al. where BSA was PEGylated and glycosylated (BSA-PEG-LA) to create ECM-like biomaterials. The crosslinked product could form a thin coating or a lyophilized 3D structure. Compared to the plastic surface of a cell culture dish, this novel material was effective in improving cell growth and proliferation more effectively than collagen coating, which had a very high degradation rate. BSA-PEG-LA reached up to five-times higher cell viability in the 3D structure compared to a plastic surface. This type of biomaterial may serve as a low-cost surface coating alternative to fibronectin or collagen [78]. Another important property of albumin is corrosion resistance. Sodium montmorillonite (MMT)/bovine serum albumin (BSA) composite coating was prepared on magnesium alloy via hydrothermal synthesis, followed by dip coating. The reason for including albumin in the composite coating is its good binding potential to polypeptides and its ability to promote cell adsorption and proliferation on the surfaces of implanted biomaterials. It was observed that the MMT-BSA coating had good corrosion resistance and better biocompatibility versus the bare Mg alloy. In vivo studies revealed that the implants with MMT-BSA maintained their structural integrity and only a slight degradation at 120 days post-implantation. These results suggest that albumin can be successfully used in its native form or as a composite coating for different biomaterials [50]. Figure 3 summarizes the most important properties of the albumin coating layer.

### 2.5. Scaffolds

Tissue damage resulting from an injury or loss of function due to aging or illness are serious challenges for regenerative medicine. Scaffolds can play an important role in new tissue formation and healing by attracting circulating cells to adhere to their surface in all three dimensions, stimulating proliferation and extracellular matrix secretion, while the slow degradation of the scaffold biomaterial makes space for new tissue formation [37]. Various materials have been developed as scaffolds, including metals, ceramics and polymers. Due to their flexibility in structural design, natural and synthetic polymers are currently the dominant scaffolding materials in tissue engineering [79]. Synthetic polymers are chosen because of their ease of processing, good mechanical properties, controllability in shape, porous structure and degradation rate. However, synthetic polymers might have lower biocompatibility and bioactivity than natural polymers [35]. Therefore, scaffold materials that can mimic the ECM are the most promising in tissue engineering, because they can provide a functional environment for appropriate cell–cell interactions, stabilize the forming tissue and serve as a reservoir of nutrients and growth factors [78]. Proteins such as silk, fibroin, fibrin, fibronectin or collagen and polysaccharides such as cellulose, starch, chitosan or hyaluronic acid are often used as ECM-like materials [80,81]. Superior mechanical properties of fibrous proteins and the prevalence of electrospinning methods for creating 3D fiber structures lead to the design of various protein-based nanofibers for scaffold fabrication [36,82]. The electrospinning of fibrous, recombinant and globular proteins was challenging for a long time, since their mechanical properties were poor. However, recent developments in material science made it possible to use globular proteins such as albumin for building three-dimensional porous scaffolds suitable for tissue engineering applications.

#### 2.5.1. Albumin-Only Scaffold

The first successful electrospinning of strong nanofibers made solely of bovine serum albumin (BSA) was achieved by Dror et al. [82]. The fabrication process involved crosslinking of the globular albumin form by opening disulfide bridges together with unfolding the protein and allowing the formation of extended structures rich with intramolecular disulfide covalent bonds. These fibers showed higher strength compared to fibers made of other natural materials; however, their biocompatibility may have been compromised [82]. There are several studies that have utilized scaffolds that solely consist of albumin achieving very good biocompatibility and stability. Nseir et al. explored the mechanical and biological features of electrospun albumin fiber scaffolds and compared them to PCL and PLLA/PLGA scaffolds. The albumin scaffolds in vivo were biodegradable (around 50% degradation after 3 weeks) and induced a mild inflammatory response when compared with implanted PLLA/PLGA and PCL fiber structures. Various cell types were successfully cultivated in vitro on vessel-like albumin scaffolds, indicating the scaffold supports cell adhesion and proliferation. Another study of this group investigated a 3D cardiac patch fabricated from albumin fibers. The expectation was that, due to the elastic nature of albumin, the fibers would not prevent cell stretching and relaxation. As a control, PCL scaffolds were utilized, since this material is already used in cardiac tissue engineering, with good results [83]. The values of the Young’s modulus of aligned and randomly oriented albumin scaffolds were close to those of the native heart ECM, while the values of the PCL scaffold were significantly higher. These results suggest that albumin patches seem to have appropriate stiffness and elasticity for incorporating them into cardiac tissues. Better water and protein absorption by albumin fibers might explain the significantly larger area of attached cells on albumin scaffolds. The authors showed that the albumin patches were able to support the assembly of functional cardiac tissues that generate strong contraction forces and have great potential in cardiac tissue regeneration [84]. Li et al. presented another processing method in which the albumin solution is converted into an albumin polymer and then freeze-dried into a solid-state tissue scaffold. The crosslinking is achieved due to the enzymatic activity of microbial transglutaminase. This albumin scaffold has a sponge-like appearance with a high water adsorption capacity and moderate mechanical strength. The surface electron microscopy analysis revealed a large pore size that supports cell attachment and proliferation. Scaffolds seeded with MSCs provided enhanced proliferation and osteogenic differentiation on the albumin scaffolds [40]. The work of Sanches et al. and the development of the AlbuCORE scaffold is another example of the successful use of an albumin-only scaffold in tissue engineering [85].

#### 2.5.2. Albumin Hybrid Scaffolds and Hydrogels

Albumin offers many advantages when applied in biomedicine; however, for some indications, it is more beneficial when albumin is combined with other biomaterials [86]. Hydrogels are hydrophilic polymers from synthetic or natural materials that have high swelling potential in the presence of water [87]. Gayet et al. proposed in 1996 a new type of hydrogel that was made by the copolymerization of bovine serum albumin and PEG [88]. The high water content (equilibrium water content > 96%) of these hydrogels allowed the controlled release of hydrophilic and hydrophobic substances, as well as small proteins. The release rates were tailored by varying the composition of the hydrogels. With a higher molecular weight of the PEG component, fewer polymer chains were bound to BSA due to steric hindrance, and the distance between two BSA molecules was greater. Therefore, the water content increased with a higher molecular weight of PEGs and defined the subsequent drug release profile. Oss-Ronen also investigated PEG-albumin, as well as PEG-fibrinogen and PEG-albumin-fibrinogen, hydrogels [89]. They confirmed the good utility of the composite hydrogels for providing sustained drug release. The release kinetics for various drugs from the hydrogel matrix were shown to be controlled by the size of the drug, the size of the PEG conjugated to the albumin and the amount of additional PEG added to the hydrogel matrix. Additionally, they evaluated the hydrogel biocompatibility by seeding them with human foreskin fibroblasts (HFFs). The PEG-fibrinogen and PEG-albumin-fibrinogen hydrogels presented good cell spreading within the matrix; however, in the case of PEG-albumin hydrogels, the cells did not spread and remained round even after 7 days of experiments, highlighting the importance of ex vivo 3D cell culture experiments, since the key biological properties could not be foreseen from the physicochemical examinations only. Another group developed biomimetic hydrogel scaffolds composed of (PEG) and collagen, fibrin or albumin. Interestingly, they also observed that PEG-albumin hydrogels exhibited poor cell spreading and migration [90]. These studies showed that hydrogel constructs can serve as a drug release device; however, the support of cell attachment and proliferation by albumin-based hydrogels was suboptimal.

Calcium phosphate-based biomaterials are frequently used in biomedicine, especially for bone regeneration, due to their similarity to the inorganic constituents of bone tissue, bioactivity, osteoconductivity and moldability [91,92]. Amorphous calcium phosphate (ACP) nanoparticles were used for the preparation of ACP-poly(d,l-lactic acid) (ACP-PLA) nanofibers with the addition of BSA and described by Fu et al. [92]. The BSA-containing ACP-PLA solution was easily electrospun into the nanofibers that displayed a fibrous structure. To stimulate mineralization, the PLA, ACP-PLA and BSA-containing ACP-PLA nanofibrous round mats were soaked in simulated body fluid with ion concentrations that are almost equal to those in human blood plasma. The morphology of the nanofibers, except for PLA, was already altered after 1 day of immersion because of the deposition of small nanoparticles on their surfaces. An extension of the mineralization time led to more intense inorganic matter deposition and the creation of a 3D nanosheet network. The surfaces of BSA-containing ACP-PLA and ACP-PLA alone became completely hydrophilic, probably due to the increased water adsorption within the newly formed porous nanosheet network. Osteoblast-like cells (MG63 cells) were cultivated on the scaffolds for 7 days, and there was a continuous increase in cell metabolic activity on all the tested fibers. Since the hybrid material with BSA and ACP-PLA achieved fast mineralization and high biocompatibility, it may have potential applications in water-soluble drug delivery systems for tissue engineering [92]. The complex dependencies between BSA and calcium were investigated by Patel et al. and Haag et al. [93,94,95]. Their modeling study revealed several conformational changes of albumin when exposed to calcium, which led to enhanced albumin bioactivity and its multilevel beneficial effects on bone tissue healing. These observations were confirmed when the polyampholyte polymer hydrogels containing conjugated albumin exposed to different calcium concentrations were seeded with MC3T3-E1 osteoblast-like cells. After 7 days, the best conditions for MC3T3-E1 cell adhesion and viability were obtained in the group with a 0.05 M calcium concentration. Interestingly, cell adhesion to the calcium-modified BSA-hydrogels was regulated by arginine-glycine-aspartic acid (RGD) and collagen-specific integrins. These results demonstrated that the delivery of bioactive calcium-modified albumin to the site of bone defect can improve cell adhesion and tissue regeneration [93,94,95].

Strategies for functional repair in the central nervous system are limited due to the low regenerative capacity of neural tissue. Nerve damage can be treated with a combination of stem cell therapies and biomaterials to promote stem cell survival on grafts and integration within the tissue, as well as local growth factors delivery to the injury site [96,97]. Hsu et al. developed a scaffold construct based on serum albumin and hemin. After the electrospinning of albumin, they doped the fibers with hemin, the oxidized form of iron protoporphyrin IX (Fe3+), which is an essential regulator of gene expression and growth promoter of cells [98,99]. The scaffolds were tested with the human episomal iPSC line and revealed good biocompatibility and support for cell attachment. A further step was to test the scaffolds’ ability for binding and releasing FGF2, which served as a model recombinant protein. It was shown that the constructs with the addition of hemin and albumin can bind FGF2 and provide a slow release profile together with enhanced proliferation of the iPS cells. Neuronal differentiation of the iPS cells was measured by the percentage of βIII-tubulin+ cells, and it was observed that hemin-doped albumin scaffolds had a higher percentage of differentiated cells. The addition of hemin provides the conductive properties of constructs, and the electrical characterization showed that, by applying voltage, a higher current passed through the hemin-doped albumin scaffolds compared to the non-doped scaffolds and PBS control. The effects of electrical stimulation on neuronal maturation and network formation were tested with the potential for neurite outgrowth and branching in the hiPSC-derived neurons. With electrical stimulation, the neurons exhibited the longest neurite outgrowth and more neurite branching on the hemin-doped albumin scaffolds compared to the other groups. The whole system was capable of incorporating and releasing growth factors to modulate cell behaviours with optimized electrical stimulation, which promoted structural maturation and enhanced neurite branching. Therefore, the hemin-doped albumin-based construct is a promising new platform for neural tissue engineering. Another example of an interesting combination of two natural biomaterials is the work of Prasopdee et al. [100]. They used bovine serum albumin and cassava starch to fabricate 3D scaffolds for liver regeneration. Similar to albumin, starch has been used as a biomaterial for tissue engineering scaffolds, bone implants, wound dressing and as a substrate for cell seeding or drug delivery systems [101]. The albumin and starch were dissolved in the colloidal solution, freeze-dried, treated with methanol or ethanol and freeze-dried again. Nontreated scaffolds after only one freeze-drying step were also included in the study. The BSA/starch scaffolds had porous structures in all three conditions, but the samples with either ethanol or methanol treatments had substantially bigger pore sizes. While testing the swelling potential, unexpectedly, the albumin/starch scaffolds without any treatment were completely dissolved in distilled water. After the alcohol treatment, the scaffolds were more resistant to water immersion and remained solid. However, the methanol treatment scaffolds had a lower swelling efficiency and lower absorption ability than the ones after ethanol treatment. A compression test showed that scaffolds with methanol treatment could tolerate a higher applied force in both a dried and hydrated state. The metabolic activity assay revealed that the viability of liver cancer cell line HepG2 for each tested scaffold was greater than 90%. Considering low costs, the natural origin of the used materials and satisfactory results, this study suggests that BSA/starch scaffolds have a good potential for use in regenerative medicine and tissue engineering. Additionally, Garcia et al. tried to avoid toxic reagents and nonhazardous reagents for the electrospinning of BSA nanofibers. They observed that the addition of ethanol (EtOH) in the solvent mixture and thermal denaturation of BSA supports the formation of nanoparticles, and the presence of hydroxypropyl methylcellulose (HPMC) favors the formation of nanofibers [102]. Albumin, a globular protein, has been important in this regard, owing to its properties such as biocompatibility, biodegradability, nonimmunogenicity, nontoxicity, water solubility, cost-effectiveness and tumor-targeting ability [7]. These studies confirmed that there is growing interest in technologies that utilize albumin as a bioactive scaffold for tissue engineering [103].

## 3. Albumin as a Local Therapeutic Agent

### 3.1. Bioactive Sites on the Albumin Chain 

One of the most important functions of albumin in clinical applications is its binding property. Albumin binds fatty acids, bilirubin, metal ions, drugs such as penicillins, sulfonamides, indole compounds, benzodiazepines, etc. [104]. There are three important molecular regions in albumin that contain cysteine and histidine imidazole residues, which are responsible for its binding capacity [26,105]. Domains IIA, IIIA and IB are binding sites for hydrophobic and hydrophilic drugs that enable the transport of substances bound noncovalently to albumin [26,106,107,108]. Oxidative stress most likely plays a significant role in the pathogenesis of sepsis, end-stage renal disease or liver failure [109], and it is believed albumin might improve patient outcomes by having a neutralizing effect [105], since cysteine-34 is a particularly redox-sensitive site of albumin [110]. Albumin has 16 histidine imidazole residues, which are major targets for a reaction with the lipid peroxidation product 4-hydroxynon-2-enal, the major product of membrane peroxidation [111] that is linked to several diseases, such as Alzheimer’s and Parkinson’s, atherosclerosis, diabetes and cancer [112]. Imidazole residues are also responsible for the buffer function of albumin [28,105]. Albumin has also the ability to stabilize therapeutic proteins, such as interferon, interleukin-2, vascular endothelial growth factor or antibodies, and improve their therapeutic efficacy [113,114,115]. Serum albumin, due to its long half-life, is prone to nonenzymatic glycation that occurs during circulation. Glycated albumin was reported as a reliable biomarker for diabetes screening and diagnosis; however, it was also shown to protect other proteins from glycation and have positive effects on neuronal and glial cells and the overall brain circulation [5,116].

### 3.2. Clinical Applications 

Albumin infusion therapy is routinely applied in patients with hypoalbuminemia; hypovolemia or shock; burns; major surgery or trauma; sepsis; cardiopulmonary bypass; acute respiratory distress syndrome; hemodialysis and the sequestration of protein-rich fluids [105,117,118], AIDS, cancer, cirrhotic ascites, erythrocyte resuspension and neonatal hemolytic disease [5,113,119]—in short, albumin has been used as a drug for severely ill patients since it became commercially available in the 1940s. Not only infusions but also albumin nanoparticles have been developed to treat cancer, diabetes, arthritis and hepatitis C [7]. However, intravenous albumin use became controversial after the publication of Cochrane Injuries Group Albumin Reviewers in 1998 and their conclusion that there is no evidence that albumin administration reduces mortality in critically ill patients with hypovolemia, burns or hypoalbuminemia and that, in these cases, it may even increase mortality [118,120]. Vincent et al. recommended cases for safe albumin use, including patients with cirrhosis, spontaneous bacterial peritonitis, septic shock and other infections, while albumin infusion should be avoided as a resuscitation fluid in patients with traumatic brain injury. Even though it is unlikely that albumin administration will have serious side effects in most patients, this therapy should be restricted to specific groups of patients in whom there is evidence of beneficial therapeutic effects [118]. Raoufinia et al. cautioned against using albumin in cardiac and renal failures, acute or chronic pancreatitis, pulmonary edema or severe anemia because of the risk of acute circulatory overload. In general, the concentration of albumin in blood over 2.5 g/dl (hyperalbuminemia) is a contraindication [5]. Serum heat-inactivation procedures are performed to prevent microbes transmission; however, these methods do not eliminate prion-mediated diseases such as Creutzfeldt-Jacob’s, although this issue might be eliminated by the use of recombinant albumin [26]. Infusions might trigger some side effects such as mild hypotension, rash, flushing, headache, urticaria, bronchospasm, fever and nausea, which normally disappear when the infusion rate is slowed or ceased [5,121]. Nonetheless, albumin infusion is indeed an important tool in critical care, and the above cases only point out that this resource has to be used only in patients with specific needs and not as a cure-all for any serious condition.

### 3.3. Albumin and Blood-Derived Products

Blood-derived products have become a standard treatment procedure over the last years. The best known and widely used products are platelet-rich plasma (PRP) and platelet-rich fibrin (PRF). These products are blood extracts obtained by processing blood at the bedside, typically through centrifugation [122]. The aim of the process is to separate the blood components in order to discard the elements considered not suitable (erythrocytes) and to concentrate the elements that may be used for therapeutic applications (fibrin, platelets, growth factors, proteins or leukocytes) [123]. Albumin is one of the main constituents of such products; however, it is rarely considered a key element in the therapeutic efficacy of blood products. There have been several reports on the successful use of blood products in orthopedic and trauma surgery, plastic surgery, spinal surgery, heart bypass surgery, chronic cutaneous and soft tissue ulcerations, wound healing and burns, periodontal and oral surgery and maxillofacial surgery [124,125,126,127,128,129,130,131,132]. Blood products are used not only in vivo but also as a regenerative factor in numerous in vitro studies. There are some limitations to PRP and PRF when using them as a replacement for a standard fetal calf/bovine serum supplement. PRP requires anticoagulants; otherwise, the cell culture medium becomes a gel and PRF is a solid fibrin membrane, so it is difficult to use it as a traditional supplement in a monolayer cell culture system. PRP and PRF are used in clinical practice as autologous products; therefore, they are relatively low-risk treatments, with the potential to improve and speed up the healing processes [133,134]. Indeed, PRP is the only therapy that has solid evidence for disease-modifying activity in knee osteoarthritis [135]. However, Magalon et al. demonstrated substantial differences among platelet-rich plasma products produced by various automated and manual protocols described in the literature [136]. These observations raise a concern that various PRP products may evoke diverse cellular reactions due to the varied contents of platelets, growth factors and leukocytes. Therefore, some studies have investigated alternatives to highly concentrated PRP products such as different types of human serums, as well as human serum albumin. In this approach, blood coagulation is induced, and the final product is a serum containing growth factors and other proteins normally present in blood, such as albumin. Serum-based blood products are cell-free, do not need any anticoagulants and remain in liquid form; therefore, their use is more convenient for in vitro systems. Among these products, we can find standard human serum albumin, autologous conditioned serum (ACS) developed by Meijer et al. [137], widely investigated hyperacute serum (hypACT), which is a highly proliferative serum-based product that might be obtained in a liquid or freeze-dried form [138,139,140,141,142,143,144,145] and plasma-derived albumin scaffold presented in several in vitro and in vivo studies by Gallego et al. [146,147,148,149], as well as low molecular weight albumin (LMWF5A) [150,151,152,153,154,155,156]. LMWF5A is manufactured by pasteurizing 5% HSA solution and filtrating it through polyvinylidene difluoride (PVDF) membranes with a 5000-Da molecular weight cutoff, resulting in a low molecular weight solution containing at least 20 different components. Except for albumin, there are other bioactive components in LMWF5A: aspartyl alanyl diketopiperazine (DA-DKP), sodium caprylate (octanoate) and N-acetyl-DL-tryptophan, which seem to have a strong influence on the LMWF5A therapeutic properties. The authors observed that LMWF5A increases the production of some anti-inflammatory factors: cytokine IL-10 and two microRNAs: miR146a and miR200b; therefore, the overall effect of LMWF5A is anti-inflammatory [153]. The clinical findings measured after LMWF5A injection into OA knee joints included pain relief, increased patient mobility and improved global self-assessment [152,154,156]; therefore, LMWF5A, together with other platelet- and serum-based blood products, seem to be in line for becoming promising therapeutics in regenerative medicine and osteoarthritis treatment.

## 4. Conclusions

Albumin might be the most universal therapeutic in the biomedical field, as it is one of the best-known proteins, easily accessible with low production costs and has excellent regenerative effects. On the one hand, with its antithrombotic, anti-inflammatory and antibacterial properties in the native form, it serves as a passivating protein for inert materials. On the other hand, if its structure is modified or albumin is combined with different biomaterials, it supports cell attachment, tissue formation and healing. Several studies investigated the clinical outcomes of albumin-based products such as BoneAlbumin, scaffolds, LMWF5A or serum-based products and reported numerous beneficial effects of these therapies. We believe that patients can benefit from including albumin-based biomaterials and therapeutics in regenerative medicine treatment strategies by achieving better clinical outcomes with minimal side effects. 

Ongoing developments indicate that albumin will be utilized in novel regenerative medicine products, taking advantage of the large library of scientific evidence for its regenerative properties and its excellent biocompatibility. As the field of regenerative medicine matures and innovations shift from the laboratory bench to the bedside, the humble albumin protein gains more recognition. More and more studies focus on explaining the complex interactions between albumin and different ligands and propose solutions for omitting the main challenge of albumin as a biomaterial, which is its relatively high degradation rate. Advanced composites using exotic nanostructured materials can have an edge in a petri dish, but once regulatory requirements enter the picture, these can hardly make it to clinical use, in some cases leaving well studies and naturally occurring proteins as the only suitable building blocks for such therapies. Serum albumin shines in these circumstances as a very malleable and robust structure protein with a good safety profile and low manufacturing cost, making it possible to move, e.g., implant coating technologies to be readily used in human therapy. We expect to see more such technologies reaching clinical development stages and ultimately appearing in the medical toolkit as albumin-based regenerative therapies.

## Figures and Tables

**Figure 1 ijms-23-10557-f001:**
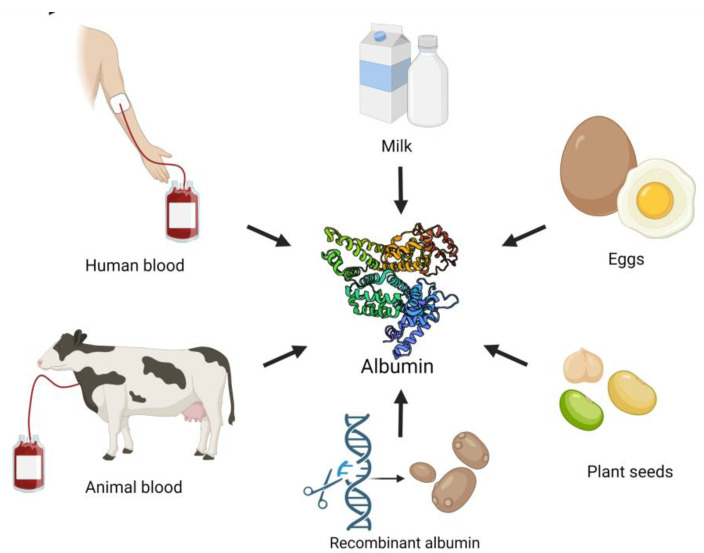
Albumin origin. Created with BioRender.com accessed on 21 August 2022.

**Figure 2 ijms-23-10557-f002:**
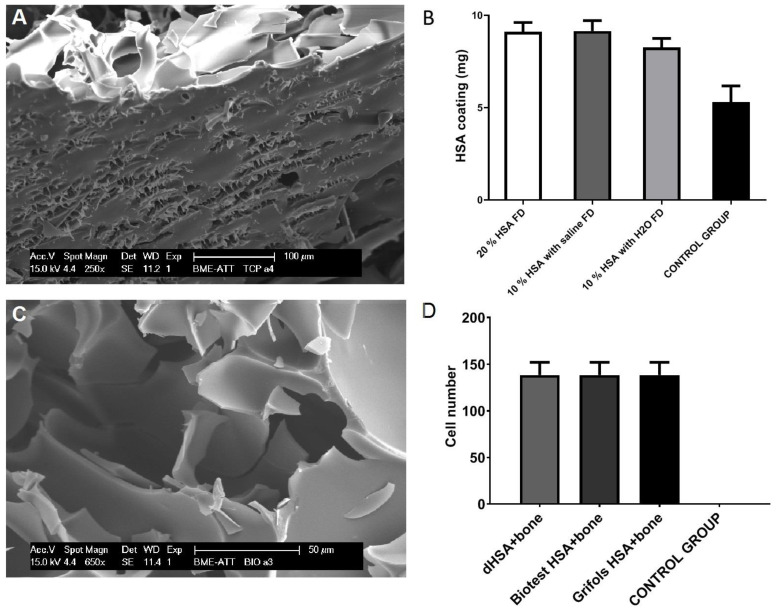
Serum albumin adsorbed onto the surfaces of mineralized bone grafts. Scanning electron microscopic images show in lower (**A**) and higher (**C**) magnification that albumin attaches to the surface as white flakes that contain ample pores and structured surfaces. (**B**) Freeze-drying of an albumin solution results in significantly higher protein adsorption than wet coating. Control group: 10% HSA with H_2_O (non-FD in the 1st step). (**D**) The number of viable MSCs attached to bone granules coated with different types of human serum albumin. There was no difference in the MSC viability after 5 days of culture on dHSA coating (delipidated recombinant human serum albumin) versus the standard human serum albumin (Biotest, Grifols). Control group: bone granules without any additives. (One-way ANOVA, *p* < 0.001). All figures are original research figures.

**Figure 3 ijms-23-10557-f003:**
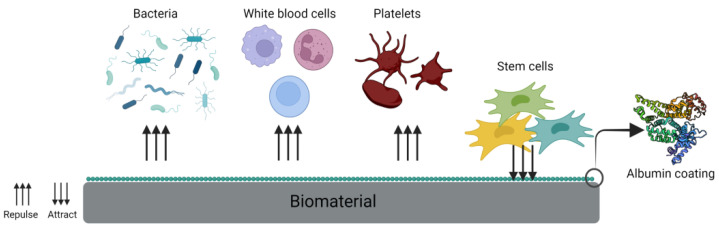
Albumin coating on a biomaterial surface. Created with BioRender.com accessed on 21 August 2022.

**Table 1 ijms-23-10557-t001:** Different albumin types and their characteristics.

Albumin Type	Molecular Weight	Structure	Main Characteristic	Production Method	Advantages	Limitations
**Human serum albumin (HSA)**	66, 5–69 kDa 585 amino acids [4]	- Non-glycosylated polypeptide [5] - Globular, heart-shaped structure - Repeating series of six helical subdomains, formed by three homologous domains (I, II, and III), each of which consists of two subdomains of similar structural motifs. [4,6]	- Acidic nature, - Stable at 60 °C till 10 h and in the pH range 4–9. - Soluble in organic solvent such as 40% ethanol [7] - Synthesized in the liver - Causes 80% of plasma colloid osmotic pressure - Binding and activation of drug conjugates. [5]	- Clinical use: - Plasma fractionation method with ethyl alcohol: Cohn method and its modifications - Purification from placenta - Heat shock fractionation - Ion exchange chromatography [5]	- Wide application: as a drug, biomaterial, in vitro supplementation or reagent - Universal and stable drug delivery career - Good availability - Low costs when used as biomaterial - Biodegradability - Lack of toxicity [5] - Cell culture supplement	- Should not be used in several clinical cases [8] - High costs of production as a drug - Relatively high degradation rate - Batch to batch variability - Risk of prion- and virus-diseases
**Bovine serum albumin (BSA)**	67 kDa 583 amino acids [4]	- HSA and BSA share 76% identity - BSA has a lower fraction of α-helices. [9]	- Physicochemical properties are similar to HSA. - More hydrophilic than HSA. - Less stable in higher temperatures than HSA. - HSA is able to crystallize and BSA does not have these properties. [4]	- Industry: - Plasma fractionation - Heat shock fractionation [10]	- Better accessibility than HSA, - Results obtained with BSA might be transferred to HSA - Universal and stable drug delivery career - Very good availability - Low cost - Ease of purification [5]	- Batch to batch variability - Risk of prion- and virus-diseases - Biosecurity risk - Non-specific binding when IgG is present in the BSA. [11]
**Ovalbumin (OVA)**	47 kDa 385 amino acids [5]	- Monomeric phosphoglyco-protein with a serpin-like structure and a helical reactive loop arrangement. [12]	- Predominant protein in albumen - Represents 54–58% of the egg white protein by weight. [12]	- Electrophoresis - Ion-exchange chromatography - Size exclusion liquid chromatography - Ultrafiltration - Adsorption, - Aqueous biphasic systems [13]	- Low cost - Availability - Can form gel networks and stabilize emulsions and foams - Good carrier for drug delivery in food matrix design - Good carrier for controlled drug release. [5]	Allergen [14]
**Lactalbumin**	14 kDa 122–123 amino acids [15]	- GIobular protein - Consists of large α-helical and small β-sheet domain, connected by a calcium binding loop [16]	- Found in bovine and human milk - Relatively heat-stable when bound to calcium and compared to other whey proteins [15,16]	- Chromatography/gel filtration - Membrane separation - Enzyme hydrolysis - Precipitation/aggregation technologies [16]	- Important source of bioactive peptides and essential amino acid - Good water solubility and heat stability [15,17] - Good results in treatment of chronic stress-induced cognitive decline; - Maintain the level of serotonin [18]	Allergen [16]
**Plant albumin**	8–16 kDa [19,20] depending on the plant source	- Usually composed of disulfide-linked low molecular weight polypeptides with a sedimentation coefficient of 2S. [21] - 2S albumins have high contents of glutamine, cysteine and methionine. [20]	- Water-soluble and highly abundant proteins - Broken-down during seed germination to provide nitrogen and sulfur for the developing seedling. - Albumins represent around 10–25% of total plant proteins. [21]	- Wet extraction methods [22]	- Possibility for so-called green preparation; - Good for foam stabilization, as it forms dense and stiff interfacial layers. [22]	Allergen [23]
**Recombinant albumin**	Structurally equivalent to HSA [24]	Structurally equivalent to HSA [24]	- Highly purified animal-, virus-, and prion-free product developed as an alternative to HSA [24] - Equally effective as HSA (Figure 2D)	- Yeast expression system [24] - Rice expression system [25]	- Possibility for so-called green preparation; - Improving cost-effectiveness and safety - No batch to batch variability [26]	- Some level of yeast-derived impurities may have the capacity toelicit a possible allergic response [24]

## Data Availability

Not applicable.

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
