# Peer review of "Albumin as a Biomaterial and Therapeutic Agent in Regenerative Medicine"

_ijms, 2022, doi:10.3390/ijms231810557_

Round 1

Reviewer 1 Report

In the present article, authors summarized the existing therapeutic uses of albumin in regenerative medicine. By considering the broad scientific interest of albumin and its unique features I recommend it for publication in IJMS after resolving the following concerns.

1.     In introduction section, authors should discuss about the importance of biocompatibility of materials in biomedicine ( cite: https://doi.org/10.1016/j.jddst.2020.102316, https://doi.org/10.1016/j.jddst.2020.102316, etc.)

2.     Figure 1, is it original research figure or adopted from literature? Cite in Figure legend if adopted.

3.     Authors should include and discuss about recent research findings on albumin-based materials in bio applications (in 2022 year).

4.     Authors should give a viewpoint at end; still what factors should consider in future biomedical applications on using albumin-based materials.

5.     Authors should discuss some details about clinical use of albumin and its importance (https://doi.org/10.2450%2F2009.0002-09)

6.     I am curious, why authors restricted to one Figure?

7.      Even though albumin having merits, but still have some disadvantages (https://doi.org/10.1186%2Fcc688). Therefore, I suggest authors should discuss the disadvantages and overcoming strategies in detail.

8.     In conclusion section, include and improve the future perspectives on albumin-based materials in regenerative medicine.

9.     Authors should carefully check some existing typos in the manuscript.

Author Response

Dear Reviewer,

The authors would like to thank you very much for reviewing our manuscript “Albumin as a biomaterial and therapeutic agent in regenerative medicine “ and considering it for resubmission.

We do highly appreciate the extensive review we received and that help us to improve our manuscript.

We analyzed and answered in detail each question and comment. Please find those answers in the reports attached below.

Additionally, we created 2 additional figures and a table with a characteristic of different albumin sources. We improved our abstract and according to the suggestion the conclusion part.

We highlighted the changes for easy tracking within the main file.

We want to once again thank you for reviewing our manuscript and hope that in the current form you will find it acceptable for publication in the International Journal of Molecular Sciences (ISSN 1422-0067) special issue "Serum Albumin in Health and Disease: From Comparative Biochemistry to Translational Medicine".

Sincerely,

Olga Kuten Pella

  1. In introduction section, authors should discuss about the importance of biocompatibility of materials in biomedicine ( cite: https://doi.org/10.1016/j.jddst.2020.102316, https://doi.org/10.1016/j.jddst.2020.102316, etc.)

Thank you for this comment. The requested modifications have been completed.

  1. Figure 1, is it original research figure or adopted from literature? Cite in Figure legend if adopted.

Thank you for this comment. Yes, these figures are our original research figures and we included this information.

  1. Authors should include and discuss about recent research findings on albumin-based materials in bio applications (in 2022 year).

Thank you for this comment. According to your suggestion, we included additionally 3 publications from Haag et al. and Patel et al. which were published in 2022. From the previous version we also have a publication from Garcia et al. We would be happy to include more recent studies, however, as we mentioned several times in our review the approach of using albumin as a biomaterial in regenerative medicine is not that widespread.

  1. Authors should give a viewpoint at end; still what factors should consider in future biomedical applications on using albumin-based materials.

Thank you for this suggestion. The requested modifications have been completed.

  1. Authors should discuss some details about clinical use of albumin and its importance (https://doi.org/10.2450%2F2009.0002-09)

Thank you for this suggestion. We included a new paragraph "3.2 Clinical applications".

  1. I am curious, why authors restricted to one Figure?

Thank you for this comment. We had a feeling it was not necessary to include more pictures, however, after this suggestion, we created 2 additional figures and added one extra panel to the previous figure.

  1. Even though albumin having merits, but still have some disadvantages (https://doi.org/10.1186%2Fcc688). Therefore, I suggest authors should discuss the disadvantages and overcoming strategies in detail.

Thank you for this suggestion. We discussed the disadvantages of albumin therapy in the new paragraph "3.2 Clinical applications".

  1. In conclusion section, include and improve the future perspectives on albumin-based materials in regenerative medicine.

Thank you for this suggestion. The requested modifications have been completed.

  1. Authors should carefully check some existing typos in the manuscript.

Thank you for this suggestion. We identified several typos and corrected them.

Reviewer 2 Report

The manuscript titled: Albumin as a biomaterial and therapeutic agent in regenerative medicine is an extensive review of the properties of albumin and how it can be used in the biomedical field. There are some points that can be improved:

The authors mention in the introduction the different origins of albumin, but depending on the origin the properties are different or do all albumins have the same properties?

It could briefly describe how it is obtained.

This molecule can be of animal and human origin, what legal and quality aspects would apply to it?

Albumin can only be used in tissue engineering. Can it be used in other advanced therapies? When you refer to regenerative medicine, what kind of therapies? It should specify what is its scope and why.

What does albumin contribute to the biomedical field as biomaterial? advantages and disadvantages compared to other biomaterials.

The conclusions must gather the main challenges to implement the use of albumin and what its role is in the biomedical field.

Author Response

Dear Reviewer,

The authors would like to thank you very much for reviewing our manuscript “Albumin as a biomaterial and therapeutic agent in regenerative medicine “ and considering it for resubmission.

We do highly appreciate the extensive review we received and that help us to improve our manuscript.

We analyzed and answered your suggestions and comments. Please find those answers in the reports attached below.

Additionally, we created 2 additional figures and a table with a characteristic of different albumin sources. We improved our abstract and according to the suggestion the conclusion part.

We highlighted the changes for easy tracking within the main file.

We want to once again thank you for reviewing our manuscript and hope that in the current form you will find it acceptable for publication in the International Journal of Molecular Sciences (ISSN 1422-0067) special issue "Serum Albumin in Health and Disease: From Comparative Biochemistry to Translational Medicine".

Sincerely,

Olga Kuten Pella

Reviewer 2

  1. The manuscript titled: Albumin as a biomaterial and therapeutic agent in regenerative medicine is an extensive review of the properties of albumin and how it can be used in the biomedical field. There are some points that can be improved:
  2. The authors mention in the introduction the different origins of albumin, but depending on the origin the properties are different or do all albumins have the same properties?
  3. It could briefly describe how it is obtained.
  4. This molecule can be of animal and human origin, what legal and quality aspects would apply to it?
  5. What does albumin contribute to the biomedical field as biomaterial? advantages and disadvantages compared to other biomaterials.

Answer to suggestions 1,2,3,4,6

Thank you for these suggestions. These comments helped us to notice that indeed, we should discuss these details about albumin (origin, main characteristic, production method, advantages and limitations). Therefore, we created a table containing all important information about different albumin sources. Additionally, we rewrote the conclusion part and summarized the advantages of albumin.  

  1. Albumin can only be used in tissue engineering. Can it be used in other advanced therapies? When you refer to regenerative medicine, what kind of therapies? It should specify what is its scope and why.

Thank you for this suggestion. We included a new paragraph "3.2 Clinical applications".

  1. The conclusions must gather the main challenges to implement the use of albumin and what its role is in the biomedical field.

                Thank you for this suggestion. The requested modifications have been completed.

Round 2

Reviewer 1 Report

Authors resolved all the concerns and now I recommend it for publication in IJMS.

Reviewer 2 Report

THe new version of review has improved.